# Modulation of the Microglial Nogo-A/NgR Signaling Pathway as a Therapeutic Target for Multiple Sclerosis

**DOI:** 10.3390/cells11233768

**Published:** 2022-11-25

**Authors:** Danica Nheu, Olivia Ellen, Sining Ye, Ezgi Ozturk, Maurice Pagnin, Stephen Kertadjaja, Paschalis Theotokis, Nikolaos Grigoriadis, Catriona McLean, Steven Petratos

**Affiliations:** 1Department of Neuroscience, Central Clinical School, Monash University, Prahran, VIC 3004, Australia; 2Laboratory of Experimental Neurology and Neuroimmunology, Department of Neurology, AHEPA University Hospital, Stilponos Kiriakides str. 1, 54636 Thessaloniki, Greece; 3Department of Anatomical Pathology, Alfred Hospital, Prahran, VIC 3004, Australia

**Keywords:** multiple sclerosis, microglia, microglial polarization, classically activated microglia, alternatively activated microglia, Nogo-A, Nogo receptor, myelin debris

## Abstract

Current therapeutics targeting chronic phases of multiple sclerosis (MS) are considerably limited in reversing the neural damage resulting from repeated inflammation and demyelination insults in the multi-focal lesions. This inflammation is propagated by the activation of microglia, the endogenous immune cell aiding in the central nervous system homeostasis. Activated microglia may transition into polarized phenotypes; namely, the classically activated proinflammatory phenotype (previously categorized as M1) and the alternatively activated anti-inflammatory phenotype (previously, M2). These transitional microglial phenotypes are dynamic states, existing as a continuum. Shifting microglial polarization to an anti-inflammatory status may be a potential therapeutic strategy that can be harnessed to limit neuroinflammation and further neurodegeneration in MS. Our research has observed that the obstruction of signaling by inhibitory myelin proteins such as myelin-associated inhibitory factor, Nogo-A, with its receptor (NgR), can regulate microglial cell function and activity in pre-clinical animal studies. Our review explores the microglial role and polarization in MS pathology. Additionally, the potential therapeutics of targeting Nogo-A/NgR cellular mechanisms on microglia migration, polarization and phagocytosis for neurorepair in MS and other demyelination diseases will be discussed.

## 1. Introduction

Multiple sclerosis (MS) is a neurodegenerative disease of the central nervous system (CNS), initiated by an autoimmune response causing severe inflammation and demyelination. The heterogeneous nature of the disease can be characterized by varying levels of demyelination, immune cell infiltration and gliosis across multifocal lesions of the CNS (see review [1]). In people living with MS, a significant health burden is observed in young adults, especially in women between the ages of 20 and 40 years, where the chronic phase is accompanied by irreversible neurological dysfunction and disability, detrimental to the individuals’ quality of life [2,3].

One of the major effectors involved in the pathogenesis of MS is an innate immune resident cell type of the CNS, the microglia. Microglia play a critical role in maintaining healthy CNS homeostasis, including regulation of synaptic development and plasticity, whilst also promoting cell survival by secreting brain-derived neurotrophic factors (BDNF) [4,5,6]. During pathogenic insults, microglial cells monitor the CNS and trigger an inflammatory response in the dysregulated CNS microenvironment, acting as the first line of the immune defense [7]. 

During an immune-mediated inflammatory cascade event in MS, microglial cells are activated (see review [8]). Through membrane polarization, microglia migrate to lesions and recruit invading peripheral macrophages, which leads to a proinflammatory response [9]. Activated microglia secrete proinflammatory cytokines IL-6, IL-12, IL-18, IL-23, interleukin-1β (IL-1β), tumor necrosis factor (TNF), chemokines CCL2, CCL3, CCL4, CCL5, CCL7 and CCL22 and reactive oxygen species (ROS) which can accelerate neuronal and oligodendrocyte cell death [10] (see review [11]). However, in a commonly used immune model of MS, the experimental autoimmune encephalomyelitis (EAE) model demonstrated remyelination may be achieved within a pathological CNS environment aiding in amelioration of disease progression, if the microglial phenotype is predominantly shifted to the anti-inflammatory alternatively activated state [12]. This alternative activation of microglia, previously designated as an ‘M2′ phenotype, is a type of polarization state in microglia and macrophages that express anti-inflammatory characteristics, compared to the classically activated (previously ‘M1′) phenotype which expresses proinflammatory characteristics [13]. Several studies on Alzheimer’s disease (AD) [14], amyotrophic lateral sclerosis (ALS) [15], Parkinson’s disease (PD) [16], epilepsy [17] and MS [18], reinforced the importance of altered microglial phenotypes during documented anti-inflammatory mechanisms that govern neurorepair, oligodendrocyte remyelination, axonal regeneration and improved cognitive and motor outcomes throughout the course of neurological disease. On the other hand, phagocytic microglia are considered fundamental in facilitating the clearance of cellular and extracellular debris, thereby modifying a vastly inhibitory microenvironment. In this context, timely removal of myelin debris may be critical in allowing oligodendrocyte migration toward demyelinated lesions, highlighting an exciting means of neurotherapeutics to enable endogenous repair through remyelination and axonal remodeling during progressive MS [19].

Nogo-A as a myelin-associated inhibitory factor (MAIF) was observed in demyelinated lesions of progressive MS [20]. The binding of Nogo-A to Nogo receptor 1(NgR1) is the strongest affinity cognate receptor to the Nogo-66 (amino acids1–40) domain of the full-length Nogo-A ligand. In this MAIF receptor union, NgR1 can trigger a downstream molecular cascade of Ras homolog family member A (RhoA) and Rho-associated coiled-coil containing protein kinase 2 (ROCK2), leading to inhibition of neuronal outgrowth and reduced synaptic plasticity [20]. In the last decade, a growing number of studies related to the effect of the Nogo/NgR signaling pathway on microglial cell function throughout various neurological disorders have emerged [12,21,22]. These studies indicated that the Nogo-A/NgR signaling pathway affects microglial cell adhesion, migration, polarization, phagocytosis and interaction with other cells in neuroinflammatory and neurodegenerative diseases. Therefore, the therapeutic mechanisms targeting Nogo-A/NgR signaling could potentially mediate microglial activation states, which may uncover novel interactions to improve neurorepair and brain neurodegeneration.

In this review, we primarily focus on the development and distribution of microglia and recapitulate the role of microglia in homeostatic mechanisms. Secondly, we summarize morphological and functional changes in aged microglia and how they affect microglial cell surveillance, phagocytosis and engagement of inflammatory responses. Lastly, we set out to define microglial activities during MS pathogenesis and explore how Nogo/NgR-dependent molecular mechanisms govern neurodegeneration and repair.

## 2. Origin, Development and Microglia Cell Homeostasis

Microglia are the key immune regulators of neurogenesis, accounting for 5–20% of neural cells [23]. They originate from the yolk sac erythromyeloid precursors, which differentiate into yolk sac macrophages, migrating and colonizing the brain during the fourth week of gestation [24,25]. Experimental validation of this is limited since the exploration of microglia via fate mapping during gestation in humans is causal. However, the identification of the microglial-specific chemokine receptor, CX_3_CR1 (observed during embryogenesis), has led to the development of the CX_3_CR1-CreER mouse model, differentiating endogenous microglia from peripherally derived macrophages [26]. Fate mapping of CX_3_CR1^GFP/+^ mice has supported this theory with the presence of CD11b^+^CX_3_CR1^+^ primitive myeloid progenitors in the yolk sac at embryonic day 8 (E8) and detected in the brains of transgenic mice as early as E9.5 [27]. The presence of these microglial cells early in the brain parenchyma correlates with neurogenesis [28,29], suggesting that microglia could play a role in early development through phagocytosis to regulate synaptic pruning and the neural progenitor cell (NPC) population [4,29]. The role of microglia in regulating the NPC population is essential for the differentiation into neurons, astrocytes and oligodendrocytes in the development of the CNS [29]. In CX_3_CR1^GFP/+^ mice, microglia were observed to regulate synapses of retinal ganglion cells at postnatal day 5 through the binding of complement component 3 (C3) and its receptor CR3, expressed by microglia. This was accompanied by increased lysosomal activity observed in this population of microglia in the dorsal lateral geniculate nucleus [30,31]. Microglia were reported to be identified in cortical proliferative zones, particularly the subventricular zone (SVZ), in rhesus monkeys at E50, where Iba^+^ (a common marker used to identify activated microglia) microglia demonstrated significant colocalized expression with the proliferating cell nuclear antigen (PCNA) [29]. As development progressed, these microglia were observed to engulf T-box brain protein 2 (Tbr2) and paired box protein (Pax6) expressing NPCs. This is suggested to lead to the reduction of neural progenitor pools during the later stages of neurogenesis, where Iba^+^ microglia were evenly distributed in the cortex with reduced pools of NPCs [29].

Microglia also secrete neurotrophic factors such as insulin-like growth factor (IGF-1), BDNF and transforming growth factor-beta (TGF-β) to support neuronal differentiation and oligodendrocyte myelination [32,33,34,35]. More specifically, observations in early postnatal rats suggest that IL-1β, interferon-gamma (IFN-γ) and IL-6 secretion from microglia promotes neurogenesis of NPCs to promote oligodendrogenesis [36]. Furthermore, the secretion of IGF-1 from these microglia promoted neuroblast migration to the rostral migratory stream from the SVZ to the olfactory bulbs, where adult neurogenesis occurs [36]. Neurotrophic support derived from microglia can also promote the expression of microglial protein such as neuropilin 1 (Nrp1), which has been reported to stimulate the release of platelet-derived growth factor receptor-α (PDGFR-α), a potent activator of enhancing oligodendrocyte precursor cells (OPCs), by potentiating their proliferation rate [37]. Studies suggest that a two-way communication relationship exists between microglia and NPCs during development, with NPCs regulating microglial cell recruitment into the stem cell niche through the release of factors such as vascular endothelial growth factor (VEGF), that in turn promote the activation and proliferation of microglia [38,39]. 

Throughout development, the morphology of microglia transitions from an amoeboid shape to an inactive phenotype that exhibits multiple fine processes. This ramified morphology supports their motility within central tissue, governing their homeostatic regulation through a unique interaction with CNS cells and their ontogenic function [40]. The homeostatic responsibility of microglia during development includes detection of alterations in the microenvironment, adequately responding to phagocytose dying cells and clearing residual myelin debris, accumulated protein aggregates and unnecessary synapses contributing to neuroplasticity [30,31,41,42].

Microglia maintain the CNS microenvironment independent of peripheral monocytes, largely due to their proliferative and self-renewal capability [43]. Proliferation rates for microglia are dependent on the area of the CNS surveyed and the technique used to assess cell cycle synthesis and mitosis rates [43,44]. Studies demonstrated that microglia have cell longevity with observed maintenance of constant populations of microglia throughout life [43,45]. However, some studies have contradicted these suggestions, indicating that while an increased number of microglial cells were observed in aged populations, decreased proliferation rates also existed [46,47]. Age is a variable that is not considered or can be controlled in many animal models of neurological diseases, and may explain the failures of translating preclinical results into clinical validation studies. Therefore, further studies are required to explain how age-dependent factors can result in microglial dysfunction and reduction in their abilities when devising novel treatments for progressive neurological diseases. 

Microglial cell markers implemented to identify their transition during pathogenesis throughout neurological decline are suggested to distinguish endogenous microglia from peripheral monocytes in human tissue studies and animal models (Table 1). 

These include specific cell membrane expression profiles identified in heterogenous populations as CD45^−^ CD11b^+^, purinergic receptor P2Y (P2RY12) and transmembrane protein 119 (TMEM119) [65,66,67,68]. However, recent studies have questioned the robust expression of TMEM119 in a cuprizone mouse model, a toxin-induced demyelination model where consumption of the copper chelator cuprizone leads to oligodendrocyte death (for review see [69]). TMEM119^+^ cells were decreased during the cuprizone-induced demyelination, whilst CX_3_CR1^+/eGFP^ cell expression increased, with less than 10% of cells demonstrating colocalization in the corpus callosum [70]. Furthermore, TMEM119 is expressed by follicular dendritic cells in the CNS; thus, its expression may not be a suitable microglial marker when labeling brain parenchymal cells [70]. Additionally, evidence suggests that microglia could lose their TMEM119 expression during the early stages of MS, implying that TMEM119 may be suited to only identify endogenous microglia in homeostatic but not pathological conditions [51]. Studies that assess the polarization of microglia also utilize markers that are expressed by peripheral macrophages, without discriminating between macrophages and microglia. These markers include CD11b^+^, Iba1^+^ and markers for specific phenotypes such as CD206 and arginase 1 (alternatively activated) expression and inducible nitric oxide synthase (iNOS; classically activated) (Figure 1) [13,71]. Thus, further research exploring polarization should utilize more specific microglial markers and incorporate single-cell RNA sequencing data to differentiate between endogenous microglia and peripheral monocytes. This would allow greater insight into the specific roles of microglia and macrophages and their impact during aging in neurodegenerative disease progression such as MS.

This line of investigation was pursued very recently by Absinta and coworkers (2021), who utilized the characteristic paramagnetic rim signatures of progressive MS lesions captured under MRI to define chronic inflammatory demyelinating edges to consist of microglia inflamed in MS (MIMS) through their specific RNA-seq profile [72]. The interrogation of the leading edges incorporating MIMS identified complement 1q (C1q) as a pathogenic driver validated through conditional deletion of this complement fragment during the course of EAE [72]. There exists an increased complexity of this profile with meningeal inflammation, a common hallmark of brain atrophy and progression correlated with CD68+ and HLA class II expression, along with a loss of P2Y2 and TMEM119 expression that can eventually promulgate a synaptopathy and neuronal loss [73]. Moreover, the metabolic profile during cholesterol metabolism of active phagocytic microglia that express TREM2 suggests that disease-associated microglia (DAMs) can regulate mTOR signaling to regulate lipid metabolism [74]. It has been recently demonstrated that TREM2 mutant microglia fail to respond effectively to lipid-rich debris (myelin cholesterol) leading to cholesterol–ester accumulation during chronic conditions of myelin phagocytosis [75]. Therefore, the evidence suggests that metabolic disruption in chronic microglial-dependent phagocytosis of myelin debris can perpetuate the neuroinflammatory state of the brain, leading to overt neurodegeneration.

## 3. Microglial Cell Function in an Aging Population

In aging, it has been postulated that aged microglia exhibit impaired phagocytosis, reduced synaptic regulation and display intracellular abnormalities in their cytoplasmic organelles [76,77,78]. Microglial cell senescence exhibits morphological changes, with retraction and ramification of microglial processes, histologically characterized by dystrophic features. Although the exact age whereby microglia switch to senescent cells is unknown, and what morphological changes lead to this is obscure, some reports suggest this could begin with instability in the cytoskeletal structure or telomere shortening, leading to metabolic dysfunction; however, further studies are required to understand what exact mechanisms are at play [79,80,81]. Nonetheless, aged microglia have been shown to exhibit reduced phagocytosis of pathological debris but paradoxically, an increase in neuronal debris phagocytosis by aged microglia was detected [82]. This finding contradicts other studies, which suggest aged microglia display an impaired ability to clear myelin debris due to a delay in the recruitment of phagocytic cells, correlated with slower remyelination [83,84]. These aged microglia were further assessed to have impaired lysosomes, limiting the capacity for myelin debris clearance. It is suggested that impaired lysosomal digestion results from oxidative stress and ROS production [85,86], leading to the accumulation of lipofuscin granules [87]. However, replacement of microglial cell populations in the experimental aged brain improved cognitive decline and restored microglial cell populations exhibiting a normal morphology, restored normal-sized lysosomes and reduced intracellular lipofuscin. Such studies confirmed that the impaired ability of senescent microglia can recover. Nonetheless, the exact trigger that drives microglial cell dysfunction in chronic neurological diseases requires further exploration [88,89].

Apart from morphological, metabolic and cell physiological changes, there exists transcriptomic changes in aged microglia. Through RNA profiling, immunoinhibitory proteins involved in microglial homeostatic regulation, such as the CD200 receptor and CX_3_CL1, have been observed to have decreased expression in aged mice [90,91]. CD200 and CX_3_CL1 are exclusively expressed by microglia in the CNS, with their corresponding ligands expressed in neurons and oligodendrocytes [92,93]. Following the removal or inhibition of these proteins, it was demonstrated that the characteristics of microglia from aged mice were restored to those exhibited in young microglia [90,94,95]. Furthermore, in individuals over 50—but not in younger adults (<30 years of age)—there was a reported increase in expression of the proinflammatory gene, secreted phosphoprotein 1 (SPP1), in microglia [96]. This change was accompanied by increased secretion of proinflammatory cytokines IL-6, IL-1β, tumor necrosis factor alpha (TNF-α) and IFN-γ by aged microglia, thus creating a subtle inflammatory environment [96]. Accumulation of excessive proinflammatory cytokines can result in sensitization of microglia and activation of inflammatory-associated pathways such as nuclear factor kappa B (NF-κB) (Figure 1) [97,98]. In contrast with an increased expression of major histocompatibility complex II (MHC-II) and CD68, the related changes in *primed* microglia may promote a proinflammatory milieu in the brain [99]. The increase in proinflammatory mediators by aged microglia is suggested to *prime* them and increase their sensitivity to the CNS microenvironment [100]. However, this creates the conundrum that whether during aging, there is reduced clearance of senescent microglia, thus leading to this chronic inflammatory state, or whether the build-up of senescent cells leads to dysfunction in phagocytosis and clearance, leading to a proinflammatory tissue milieu. Nonetheless, these characteristics and accumulation of low-grade inflammation driven by senescent microglia, suggest that in an aged CNS, microglia may have decreased neuroprotective effects and can increase the susceptibility and propagation of neurodegenerative diseases such as MS.

## 4. The Role of Microglia in Propagating Neuroinflammation and Neurodegeneration in Multiple Sclerosis

The most common lesions present in MS individuals with short disease duration exhibit partial or ongoing demyelination with microglial cell activation [9,101,102]. These lesions can develop into hybrid forms of active and inactive lesions known as mixed or chronic lesions. These lesions are more abundant in patients with a disease duration of around 10 years and most common in people living with progressive MS [102]. Microglia have a decreased presence in the lesion core but are abundant around the peri-plaque rims [1]. Within these inflamed rims, these activated microglia have been categorized by Absinta et al. (2021) into two distinct populations. One population is observed to depict a foamy appearance with upregulation in lipid phagocytosing genes, whilst the second population is suggested to be involved in antigen presentation and complement protein activation [72]. The microglia in the lesions are accompanied by ongoing myelin loss, thus giving it a smoldering appearance [101]. 

In patients who have had MS longer than 15 years, predominantly secondary progressive MS (SPMS) patients, the lesions are mainly inactive and have an absence of microglial cells with extensive demyelination. Interestingly, the presence of microglia in these lesions was observed to be less than in healthy individuals [101]. In SPMS post-mortem brain tissues, lymphoid cell follicles which are composed of antigen-specific plasma cells entangled with reactive T cells and follicular dendritic cells were observed, which is similar to the microenvironment of the germinal center in secondary lymphoid organs (Figure 2) [103,104,105]. These B cell follicle-like lesions can be found in the brain parenchyma but more frequently appear in the cortical meninges and perivascular spaces of subcortical white matter in frontal, temporal and parietal lobes and in the cingulate gyrus of SPMS brain tissue [103,104,105]. The specific distribution of B cell follicle-like lesions may be clinically related to significant neurological and behavioral dysfunction in progressive MS patients [103]. Interestingly, these findings identify enhanced stimulation of humoral immune responses in progressive MS patients, and therefore could potentially exacerbate the chronicity of MS, especially in aged populations [103,104,105].

Moreover, activated B cells persist in lesions and meningeal aggregates, and are considered to be associated with microglial activation and grey matter demyelination [103]. It is of note that more evidence illustrated that the crosstalk between microglia and activated B cells is bidirectional. Touil et al. (2017) reported that classically activated microglia have an upregulated expression of CD86 in B cells. Conversely, activated B cells reduced anti-inflammatory IL-10 secretion from microglia and macrophages, which could possibly regulate microglial phenotypes and functions during progressive stages of MS [106].

Apart from the cross between activated B cells and microglia, the production of insulin-like growth factor 1 (IGF-1) by microglia induces the proliferation of astrocytes and can promote glial scar formation, predominantly observed in chronic active lesions (Figure 2) [107]. These are the result of astrogliosis with reactive astrocytes, microglia and NG2^+^ glia, which respond to create a barrier that surrounds the injured area [107,108,109]. In an animal model of MS, these glial scars were shown to prevent perivascular leukocytes from invading the CNS and transgenic ablation of the scar-forming astrocytes, which increased inflammation and the clinical course [110]. While the formation of glial scars at the site of injury may be beneficial at first to reduce inflammatory responses and damage, glial scars may also act as a barrier to axon regeneration and remyelination. In glial scars, reactive astrocytes produce extracellular matrix proteins such as chondroitin sulfate proteoglycans, which inhibit OPC differentiation, remyelination and axonal regrowth [111] (for review see [112]). Microglia and astrocytes are also capable of pruning axon synapses, which if done excessively, can disrupt neuronal connectivity (for review see [113,114]). Thus, breaking down these scars in the center of a lesion and removing the extracellular matrix is crucial for neurorepair, with the role of microglia central to inflammation and degeneration observed in progressive stages of MS, presenting itself as a potential target for neurorepair.

Microglia may play an integral role in initiating repair in MS and its respective animal models, suggesting its operation by eliciting an anti-inflammatory response, enhancing the capability of inhibitory myelin debris clearance and stimulating OPCs to remyelinate (see review [115]). This was observed in another toxin demyelination animal model, the lysolecithin animal model, which was used to study factors that may influence demyelination and remyelination during MS [116]. A decrease in motility and phagocytic activity of myelin debris was exhibited by the presence of CX_3_CR1^GFP+^ microglial cells in demyelinating lesions of aged (9 to 12-month-old) when compared to young (2 to 3-month-old) lysolecithin treated mice, identified through multiphoton live cell imaging [117]. Inefficient clearance of myelin debris by microglia has been shown to hinder remyelination, by limiting OPC differentiation or suppressing phagocytosis of inhibitory MAIF expressed in myelin debris [118]. Interestingly, deficiency of CX_3_CR1 in young mice generates a transcriptional profile similar to aged mice [119]. Furthermore, lysolecithin injection in older mice leads to increased lesions with more profound axonal loss and demyelination [120], thus highlighting the importance of microglia during remyelination.

Environmental stimuli binding to receptors can activate resting microglia into polarized states, playing a large role in the function of microglia to phagocytose and remyelinate during neurodegenerative disease. Classically active M1 microglia are potentiated in the presence of IFN-γ or lipopolysaccharides (LPS) in the tissue environment, binding to microglia surface receptors such as toll-like receptor 4 (TLR4R), IL-1R and TNF receptor, which is suggested to trigger intracellular signaling cascades, leading to activation of proinflammatory mediators such as NF-κB (Figure 1) (See detailed review [121,122]). This, in turn, stimulates the release of proinflammatory cytokines and chemokines (TNF-α, IL-6, IL-12, IL-1β) whilst downregulating IL-10 [123] and activating iNOS to produce ROS (75). Contrastingly, the polarization of microglia to the alternatively activated ‘M2′ phenotype can be induced in the presence of IL-4, IL-10 or IL-13 binding to their respective receptors on the surface of microglial cells [124,125,126]. Upon binding, the JAK/STAT pathway is triggered, leading to the upregulation of anti-inflammatory markers CD206, IL-10, TGF-β and arginase 1 in activated microglial cells, along with a coordinated T-helper 2-cell response (see detailed review [127,128,129]). These mechanistic observations suggest that microglial cell activity can be regulated by cytokine-dependent polarization, leading to the expression of phenotype-specific markers. 

This type of polarization has been observed in MS lesions, with microglia having an intermediate phenotype, possessing both classically activated and alternatively activated markers [18]. It has been reported that in chronic active lesions, increased markers of classically activated microglial phenotypes exist, alongside diminished numbers of macrophages and microglia that express the markers commonly associate with alternatively activated phenotypes. These data have suggested that there may be a slow decline in potential neuroprotective microglial cell numbers where active lesions transition into chronic progressive neurodegenerative lesions [18,60,130]. Chronic lesions are prominent in patients with protracted disease duration and are likely observed as aged lesions, consistent with the observation in older experimental mice, where higher levels of proinflammatory factors, such as IL-1β and TNF-α, in coordination with elevated classically activated phenotypically defined microglia could be identified. Additionally, reduced expression of arginase 1, a marker for alternatively activated microglia, was identified following tibial fracture surgery [61]. The data contribute to a clearer understanding of sustained low-grade inflammatory milieu created by senescent and aged microglia that could contribute to the persistent classically activated microglial senescent ability, leading to chronic stages of the disease with neurological progression (Figure 2).

Thus, a shift towards a predominantly alternative-activated phenotype in microglial populations could be crucial in limiting chronic neural inflammation and directing a degenerative inhibitory environment to a repair environment, which may provide novel therapeutic targets for neurodegenerative disease. In a model with an ex vivo administration of alternatively activated microglia directly into the CNS of EAE-induced mice, disease progression was found to be suppressed or ameliorated [131]. Whereas the shift from classically activated towards alternatively activated also coincided with enhanced recruitment and differentiation of OPCs required for remyelination, the removal of alternatively activated microglia led to deficient OPC recruitment [60,132]. This finding is in line with another study that described the presence of alternatively activated microglia within inactive lesions and rims of chronic active lesions to show the greatest remyelination [60]. This is further supported by increased secretion of neurotrophic factors (NGF, BDNF, IGF-1) from alternatively activated microglia, which improve OPC proliferation and differentiation [9]. In addition, increased phagocytosis of remyelination inhibitors and myelin debris within TREM2^+^ alternatively activated microglia highlights the importance of this microglia phenotype in contributing to remyelination and repair [118,132].

## 5. Overview of Nogo-A and Nogo Receptor 1 Signaling Pathway in Neurodegenerative Disease—A Potential Role in Microglial Activation

Nogo-A, along with Nogo-B and Nogo-C, is encoded by the reticulon gene 4 (RTN4) with translated proteins sharing a common 188 homologous amino acid C-terminus. However, Nogo-A, -B and -C proteins display variable expression in different cell types and have distinct and independent functions [133,134]. Nogo-A is localized to the CNS and is important in regulating neurite outgrowth [135]. Nogo-66, a hydrophilic loop located on the extracellular domain of the C-terminus of Nogo-A, has nanomolar affinity binding for its cognate receptor, NgR1 [136,137,138]. Despite this high-affinity ligand interaction, NgR1 lacks a cytoplasmic domain capable of binding and activating adaptor proteins to initiate downstream intracellular signaling. Thus, NgR with its co-receptors immunoglobulin-like domain-containing protein 1 (LINGO-1), and the lower affinity neurotrophin receptor p75 (p75^NTR^) or tumor necrosis factor receptor superfamily member 19 (TROY), are required to heterodimerize in a receptor complex consisting of NgR/LINGO1/p75 or NgR/LINGO1/TROY, potentiating signaling that elicits cytoskeletal reorganization [135,139,140]. Once Nogo-A activates the NgR1 complex, RhoA-GTP is phosphorylated to ROCK2, which in turn can phosphorylate the collapsin response mediator protein 2 (CRMP2) at the threonine 555 amino acid site; demonstrated to promote axonal retraction and potentiate axonal sprouting observed in neurological disease models [20]. The stimulation of Nogo-A/NgR/RhoA/ROCK/CRMP2 may lead to axonal microtubule disassembly and/or actin depolymerization through the inhibition of Rac-GTP activity and cofilin phosphorylation [20].

Over-expression of Nogo and its cognate receptor complex has been seen in hippocampal tissue sections and in senile plaques of AD patients [141]. Additionally, both NgR1 and Nogo-A can be detected in chronic MS lesions, suggesting the involvement of Nogo-A and its receptor in the pathophysiology of MS [136,142,143]. Our group has identified that following EAE induction, axonal damage is associated with an increase in NgR1-dependent phosphorylation of CRMP-2 that may be observed in archived brain and spinal cord tissue with chronic active demyelinating MS lesions [144]. In the last decade, the Nogo-A/NgR/RhoA/ROCK signaling pathway has been reported as the active molecular mechanism that drives axonal guidance in severe and long-term functional deficits within the adult CNS [134,138,145,146]. Moreover, it is well documented that activation of the Nogo-A/NgR-dependent signaling pathway can promote axonal degeneration in AD, MS and spinal cord injury (SCI) [136,137]. Interestingly, during spinal cord neurodegenerative disease, in the case of the transgenic SOD1^G86R^ model of ALS, Dupuis et al. (2002) first demonstrated that the level of Nogo-A increased at early stages of the disease [147]. Jokic et al. (2006) then crossed SOD1^G86R^ mutant mice with Nogo-A^−/−^ mice and demonstrated that the SOD^G86R^/Nogo-A^−/−^ mice had longer survival times and decreased muscle denervation compared to the control group in the ALS transgenic model [148]. However, Yang et al. (2009) later observed that Nogo-A affected the redistribution of protein disulfide isomerase (PDI) to enhance survival in SOD1^G93A^ ALS transgenic mice [149]. It could be speculated that the discrepancy in their findings may be related to the different SOD1 mutations expressed in these transgenic mice with aberrant copper metabolism central to the pathogenesis, driving neurodegeneration in the former transgenic model. 

Inhibition of the Nogo-A/NgR signaling pathway has resulted in axonal regeneration and functional recovery in animal models of SCI [150,151]. Similarly, blocking the Nogo-A/NgR pathway demonstrated enhanced axonal plasticity and behavioral improvement in the rodent model of stroke [152,153]. A study by Rust et al. (2019) also discovered that Nogo-A^−/−^ and S1PR2^−/−^ mice with cerebral ischemia were observed to have improved vascular repair compared to their wild-type counterparts [154]. These results provide new pathophysiological insights when attempting to target the Nogo-A/NgR signaling pathway during chronic neurological disorders.

In the transgenic model of AD with increasing amyloid beta burden, the expression of NgR1 on reactive microglia in APP/PS1 transgenic mice was identified, which was correlated to NF-κB/STAT3 activation in these reactive cells during inflammatory progression [155]. This suggested that the Nogo-A/NgR signaling pathway within the degenerative CNS is more complex than the molecular events elicited during axonal degeneration, but is also implicated in microglial activation propagating neuro-inflammation. Consistently, in progressive MS post-mortem brain tissues, NgR1 was strongly expressed in astrocytes and in more than 60% of microglia at the site of chronic active and inactive demyelinating injury, with these glial cells also expressing both of the putative NgR-dependent co-receptors, TROY and LINGO 1 [142,143,156]. The preliminary data obtained from our laboratory clearly indicate the co-localization between CD206^+^ and CD68^+^ microglia with Nogo-A, respectively, in various neurological diseases such as AD, frontotemporal dementia (FTD) and MS. This significant finding allows us to postulate that microglial phagocytosis of Nogo-A myelin debris can be regulated by the Nogo-A/NgR signaling pathway in microglia. Collectively, the question of whether the function and activities of microglia can be affected by the Nogo-A/NgR signaling pathway has arisen, which may be a potential mechanism that can be targeted to achieve the modulation of proinflammatory microglial activity and potentiate functional recovery in numerous neuroinflammatory and neurodegenerative disorders, is yet to be posited.

## 6. Effects of Nogo-A/NgR1 on Microglia Activity

Recently, several studies using animal models of MS, AD, SCI and traumatic brain injury (TBI) have set out to examine whether microglial cell function and activity are affected by the Nogo-A/NgR signaling pathway. Hereafter, we discuss recent findings related to the Nogo-A/NgR signaling pathway in microglia migration, polarization and phagocytosis, and evaluate the clinical efficacy of targeting Nogo-A/NgR to improve the activated status of microglial cells in varying neurological diseases of different etiology.

### 6.1. Microglia Migration

The diverse range of cytokines, chemokines and extracellular matrix substrates play a role during CNS disorders, with the inhibitory endogenous micro-environment surrounding axonal degenerative lesions previously established to be repulsive for myelination during acute and chronic inflammatory responses (see review [157]). Activation of integrin receptors collaborates with the downstream Rho family guanosine triphosphate (GTP)-ases Rac1, Rho and CDC42 to mediate microglial mobilization by surveying filopodial protrusions and establishing adhesion formation and cell-body contraction through cytoskeletal reorganization [158,159]. It has been speculated that microglial cell migration may be related to the regulation from upstream Nogo-A/NgR signal stimulation.

In a transgenic model of AD, active knockdown of NgR displayed a reduction in astrocyte and microglial cell numbers within the hippocampus, suggesting that NgR1 may be involved in microglial cell recruitment and in active migration [160]. Through migration assays performed in cell culture, it was demonstrated that Nogo-66 can suppress microglial cell adhesion of NgR1 whilst limiting their migration [161]. Furthermore, Nogo-66 resulted in the aggregation of microglial cells without enabling their polarization and protrusions, governed through inhibition of the cytoskeleton rearrangement [161]. On a molecular level, this study found that RhoA was activated by Nogo-66 but the expression of CDC42 was reduced, providing evidence that microglial cell migration is inhibited when the Nogo-A/NgR downstream signaling mechanism is operative [161]. In another model for AD, 15-month-old APP/PS1 transgenic mice showed diminished capability of microglial cell migration to amyloid beta (Aβ) fibril deposition mediated through the Nogo-A/NgR-dependent mechanism [21], indicating that in the aged brain, Nogo-A/NgR stimulation can affect inflammatory mechanisms within the CNS. Conversely, obstruction of Nogo-A/NgR signaling was shown to contribute to elevated microglial cell recruitment towards amyloid plaques and increased the expression of CD36 on these activated microglial cells [21]. These results identify that aging could result in altered activation of the Nogo-A/NgR-dependent intracellular signaling pathways in microglia, which contribute to decreased microglial migration in the chronic neurodegenerative environment for efficient amyloid clearance, thereby contributing to the amyloid burden in the brain. This could be explained by the interaction of NgR1 on microglia with newly synthesized myelin containing Nogo-A, myelin oligodendrocyte glycoprotein (MOG) and myelin-associated glycoprotein (MAG) ligands; thus, leading to potential inhibition of resting microglia to become activated motile cells able to contribute to local neuroprotection [162]. However, the function of these microglia with deficient motility needs to be further investigated. 

Interestingly, another study performed in aging mice following TBI, reported higher expression levels of NgR1 and TROY on microglia compared to young mice in regions of the cerebral cortex and basal forebrain, but no significant changes were observed within the hippocampus [163]. It is recognized that aging affects the elevation of NgR1 levels on microglia which shows the specific spatiotemporal distribution in CNS injuries, which not only increase the tendency of microglial cells to migrate but also exerts directional orientation for their migration to specific regions within pathological lesions.

The Nogo-A/NgR paradigm has been suggested as a potential mechanism involved in multicellular interactions linked to inflammation, demyelination and axonal degeneration. However, to our knowledge, studies have failed to explore and discuss how Nogo-A/NgR signaling pathways can mediate the interaction of microglia with other CNS cells during neurodegeneration. Future studies should be designed to understand the clinical implication of microglia and attempt to answer the question of whether there is an interaction between Nogo-A/NgR and microglia and other intrinsic neural cells. Whether this mechanism is related to promoting remyelination and neuro-regeneration requires investigation. Recently, cellular therapeutics targeting Nogo-A/NgR, such as NgR-Fc fusion protein, have shown significant recovery in primate models with SCI and have now commenced a phase 1 clinical trial [164] (NCT03989440). Nonetheless, to achieve the jump to clinical translation, further investigation of its neurological and therapeutic effect is required, to target the unmet medical need for neurorepair prominent in diseases that include progressive MS.

### 6.2. Microglia Phagocytosis

Various receptors are expressed by microglia to facilitate phagocytosis, including toll-like receptors (TLRs), triggering receptors expressed on myeloid cells 2 (TREM-2), Fcγ receptors (FcγR), complement receptors, scavenger receptors and the MerTK family of receptors (see reviews [42,165,166]). However, the activation of NgR1 expressed on microglia may impair this phagocytic activity.

A recent study performed in the APP/PS1 transgenic mouse model of AD has uncovered that aged microglia express a higher level of NgR1 compared to young microglia, and these aged microglia exhibit a decreased ability to clear Aβ deposits [22]. The decrease in microglial cell Aβ phagocytosis was identified to be mediated through the activation of the NgR1-ROCK-Smad2/3 signaling pathway, which lowered the expression of CD36, a scavenger receptor for phagocytosing Aβ [22]. In MS, a variety of receptors are involved in the process of myelin clearance by microglia such as CR3, scavenger receptor AI/II (SRAI/II), TREM2, MerTK and CX_3_CR1 receptors [118,167,168,169]. Gitik et al. (2010) observed that Rho/ROCK signaling downregulated the CR3-mediated phagocytosis of inactivate C3b (C3bi)-opsonized and non-opsonized degenerated myelin, whereas myosin light chain kinase (MLCK) activated the phagocytosis activity [170]. This suggested that phagocytic mechanisms governing primary microglia may be regulated by the rearrangement of the cytoskeleton. Therefore, activation of Rho/ROCK via Nogo-A/NgR-dependent signaling may decrease microglia phagocytosis elicited through the CR3 receptor. However, Scheiblich and Bicker (2017) found that inhibition of RhoA/ROCK decreased microglial phagocytosis activity in vitro [171]. The discrepancy may be due to the different types of particles being phagocytosed or the receptors utilized to explore microglia phagocytosis. Interestingly, different approaches were taken by Gitik et al. (2010), who performed cell culture assays utilizing degenerated myelin as the target of phagocytosis, whereas Scheiblich and Bicker (2017) used neuronal cell fragments.

Resident microglia, especially in their alternatively activated state, were shown to engulf myelin more efficiently than peripheral macrophages [172] with reduced susceptibility to cell death following the phagocytosis of myelin [173]. Lampron et al. (2015) demonstrated that insufficient removal of myelin debris in the corpus callosum by microglia in the cuprizone-induced demyelination model hindered remyelination of axons [118]. However, they only investigated the CX_3_CR1-dependent phagocytosis mechanism in microglia, and they did not differentiate between microglia and macrophages. Alreheili et al. (2018) found that there was a greater extent of myelin debris phagocytosis by microglia in NgR1 knockout mice [12]. Nevertheless, the specific physiological role of the NgR homologs NgR1 and NgR3 in microglial cell myelin debris phagocytosis were not studied. Since inhibition of the Nogo-A/NgR signaling pathway in microglia has been shown to enhance the clearance of harmful substrates in the CNS, antagonizing the Nogo-A/NgR pathway and improving the phagocytic capacity of activated microglia can be used as a therapeutic strategy to ameliorate neurodegenerative diseases.

### 6.3. Microglia Polarization

The role of NgR1 during specific neuroinflammation states is unknown, but what needs to be determined is whether NgR1-dependent signaling in the microglial cells may outlast polarized phenotype activity, or whether the neurological polarization is governed by neighboring neural cell expression. In this same line of investigation, it was identified that an increase in expression of NgR1 on activated microglia and macrophages was reported to be co-expressed with the alternatively activated phenotype marker, Arginase 1, but not the classically activated iNOS^+^ phenotype microglia in spinal cord lesions from mice in the acute and chronic phases of EAE [156]. This data suggests that NgR could play a role in the polarization of microglia; however, this study interrogated both microglia and macrophages; thus, future studies solely exploring the expression of microglia could elucidate the role of NgR in governing the microglial cell activation status. Similarly, Arehalli et al. (2018) identified that in NgR1 mutant mice, there was a decreased severity and a delayed onset of EAE, with a substantial shift towards the alternatively activated status in microglia/macrophages [12]. Thus, this provides a potential target to limit NgR1 function in microglial cells that may phenotypically modify their physiological role to alleviate disease progression in chronic inflammatory diseases such as MS.

Amelioration of disease, orchestrated through shifting the balance towards an alternative activation status in microglia, has been observed in other neurodegenerative diseases such as AD [174]. NgR expression is upregulated during AD, suggesting that neuroinflammation through this pathway via NF-κB activation can lead to an unbalanced proportion of proinflammatory and anti-inflammatory microglia [175]. The formation of Aβ aggregates can activate microglia towards an M1 phenotype with increased production of proinflammatory cytokines such as IL-1β, TNF-α and IL-6 [176]. Furthermore, studies have suggested that Aβ can activate NF-κB in astrocytes and, in coordination with microglial activation and polarization, can promulgate proinflammatory mechanisms in neurodegeneration [26].

The downstream activation of NF-κB could be initiated by ROCK activation via the AT1 receptor on microglia, shifting the activated status to the classical pathway [177]. ROCK II, has been demonstrated to increase the expression of CD206 in the hippocampus of the transgenic mouse model of AD. Furthermore, ROCK inhibition can lead to a decreased microglial cell production of IL-1β and TNF-α, which is commonly produced by microglia engaged during the proinflammatory classical pathway [178]. The blockade of NgR1 in APP/PS1 transgenic mice was observed to result in an increase in alternatively activated markers such as Arg1, CD206, IL-4 and IL-10 [178], correlating with decreased extracellular Aβ deposition [176]. Similarly, this was demonstrated in the transgenic ALS mouse model, where ROCK inhibition of microglia in the SOD1G93A mutant mouse spinal cord displayed proinflammatory cytokine and chemokine profiles, including elevated TNF-α, IL-6, CCL3 and CXCL1 expression, accompanied with improved motor function [79].

Despite the exact signaling pathway that leads to microglial cell polarization being undetermined, it undeniably contributes to multiple signaling pathways. As previously defined, inhibition of NF-κB activation can lead to the downregulation of inflammatory genes, switching the microglial status to the alternative-activated phenotype [179]. Studies have suggested that activation of the microglial TLR4 through the NF-κB p65 protein leads to the polarization of microglia towards the classical-activated pathway phenotype [180]. Translocation of NF-κB p65 by the ROCK2 inhibitor, Fasudil, has also been shown to shift from a classically activated to an alternately activated microglial phenotype, suggesting an association with the Rho/ROCK signaling pathway [179,181]. These pathways have been associated with TLR4 stimulation and may also initiate the MAPK signaling pathway, which leads to the upregulation of proinflammatory genes [182]. STAT signaling pathways also influence polarization with STAT1 and STAT2 activation inducing the release of proinflammatory cytokines and chemokines, whereas STAT3 is shown to promote the activation of alternatively activated microglial cells [183,184,185,186]. Nonetheless, further studies are required to elucidate the pathways that effectively shift the phenotype of microglia towards an anti-inflammatory phenotype.

## 7. Conclusions

Although MS is commonly diagnosed in women aged between 20 and 40 [187], the highest mortality rate is still found in people around 50–60 years old during the protracted chronic neurodegenerative stage, where the burden of disease is the greatest [188]. Therefore, the urgent unmet medical need must be addressed through the design of therapeutics targeting the population of people living with long-term progressive MS. Microglia act as resident immune cells maintaining CNS homeostasis and are activated when pathogens invade the protected endogenous environment. However, aging can lead to the elevation in activated microglia with proinflammatory phenotypic signatures contributing to the formation of a chronic inflammatory microenvironment. In MS, the function of microglial morphological and phenotypic profiles can be altered significantly during the evolution of pathological lesions. Several studies have indicated that during the chronic stage of MS disease, microglia with inactive lesions show decreased activity, enabling myelin debris engulfment, along with the increased presence of the classical pathway phenotype, remyelination and axonal regeneration.

Considering the dynamic role of microglia in the pathology and pathogenesis of chronic MS, it is important to identify the microglial cell dynamics and the underlying mechanisms they serve during neurodegeneration and repair. Moreover, the therapeutic option to target the modulation of microglia has emerged in recent years; however, the fundamental pathological mechanisms remain elusive but may be key to designing cellular therapeutics, enabling clinical recovery for progressive MS patients. Microglial cell migration, engulfment and phenotypic changes affected by the Nogo-A/NgR signaling pathway may provide a novel strategy for enhancing neurorepair and neuro-regeneration during chronic neurological deficits. Obstruction of the Nogo-A/NgR signaling pathway along with the downstream activation of RhoA/ROCK can result in elevated microglial phagocytosis of biological neural debris and enable the phenotypic shift from classically activated to alternatively activated polarization states, demonstrated throughout pre-clinical studies of AD and MS.

## Figures and Tables

**Figure 1 cells-11-03768-f001:**
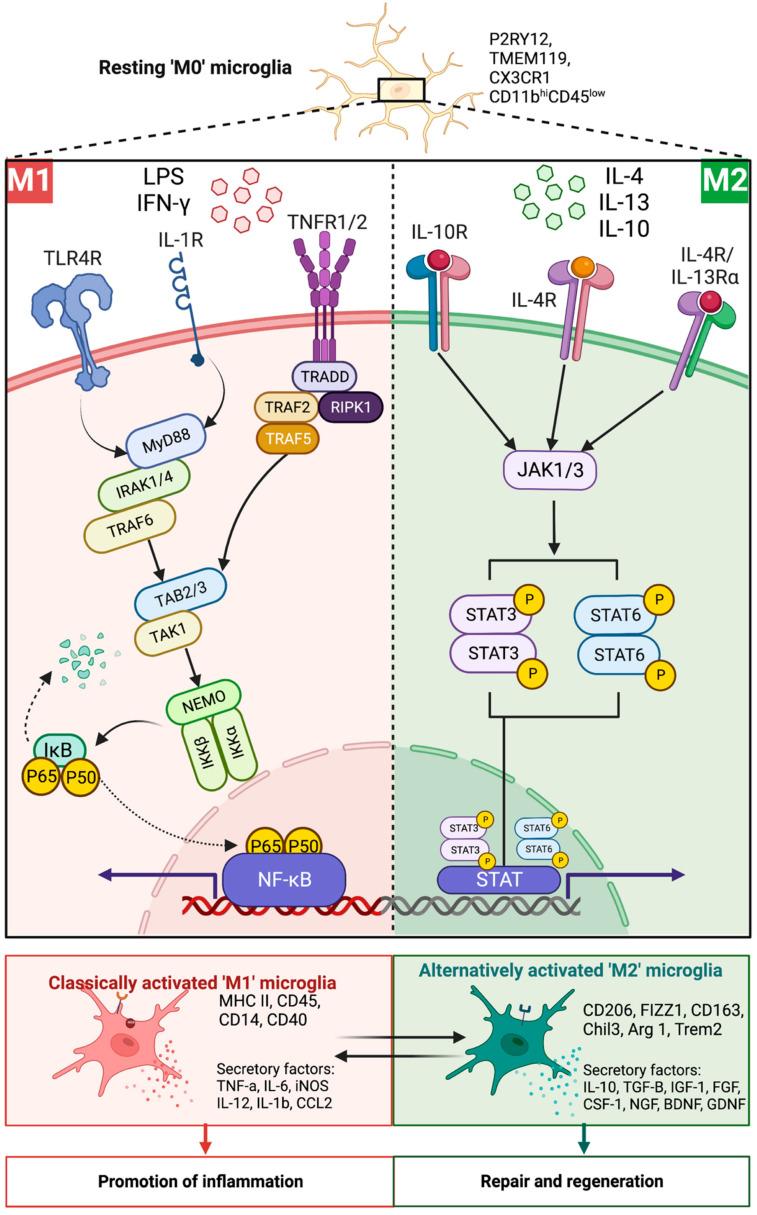
The mechanistic activation of classically activated ‘M1′ and alternatively activated ‘M2′ microglia. In the presence of interferon-gamma (IFN-γ) and lipopolysaccharides (LPS), the resting ‘M0′ microglia undergo polarization towards the classically activated ‘M1′ proinflammatory phenotype by activation of the nuclear factor kappa beta (NF-κB)-dependent pathway. In contrast, the IL-4, IL-10 and IL-13 cytokines trigger signal transducer and activator of transcription (STAT)3/STAT6 phosphorylation within the resting microglia, transitioning cells into the alternatively activated phenotype and increasing the translation of anti-inflammatory cytokines. Under proinflammatory conditions, the classically activated ‘M1′ microglia exacerbate neuroinflammation, whilst the alternatively activated ‘M2′ phenotype can promote repair and regeneration. BDNF: Brain-derived neurotrophic factor. Chil3: Chitinase-like protein 3. CSF-1: Macrophage colony-stimulating factor-1. FGF: Fibroblast growth factors. FIZZ: Found in inflammatory zone. GDNF: Glial cell-derived neurotrophic factor. IGF: Insulin-like growth factor. IKK: I kappa β kinase. IRAK: Interleukin-1 receptor-associated kinase 1. JAK: Janus kinase. MHC: Major histocompatibility complex. MYD88: Myeloid differentiation primary response 88. NEMO: Nuclear kappa- β essential modulator. NGF: Nerve growth factor. RIPK1: Receptor-interacting serine/threonine protein kinase 1. TAB: Transforming growth factor β activated kinase 1 and MAP3K7 binding protein 2. TAK1: Transforming growth factor β activated kinase 1. TGF-β: Transforming growth factor β. TLR: Toll-like receptor. TNF: Tumor necrosis factor. TRADD: Tumor necrosis factor receptor type 1-associated DEATH domain protein. TRAF: Tumor necrosis factor receptor-associated factor. TREM2: Triggering receptor expressed on myeloid cells 2. [Illustration created in BioRender.com accessed on 17 October 2022].

**Figure 2 cells-11-03768-f002:**
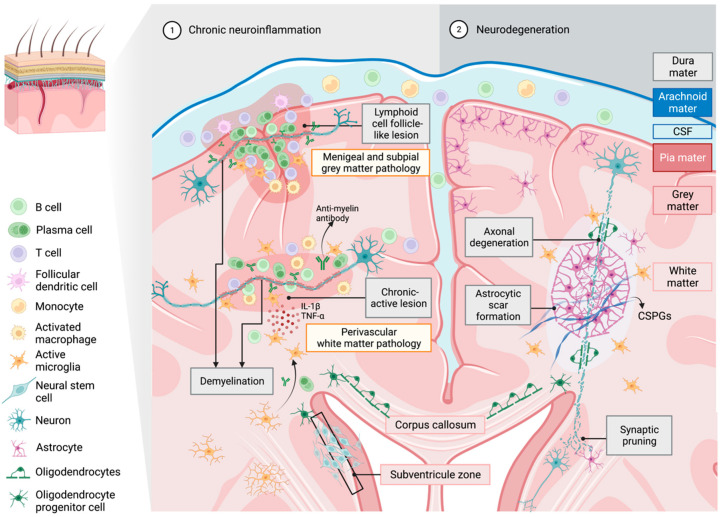
Microglia during chronic neuroinflammatory and neurodegenerative conditions. B cells with other lymphocytes and myeloid cells residing in the cerebrospinal fluid (CSF) infiltrate the subpial and meningeal cerebral cortex and the region of perivascular spaces, forming ectopic B cell follicle-like lesions as observed in progressive multiple sclerosis. As these cells percolate through the cortices towards the axon, resulting in chronic inflammation in grey and white matter, microglia that are activated by bound myelin-specific immunoglobulins secrete proinflammatory cytokines, contributing to promulgated myelin and axonal damage. Under neurodegenerative conditions, the disrupted myelin sheath can act as one of the main precursors for further axonal damage, followed by the formation of reactive astrocytic scarring potentiated through the interaction with activated microglia. [Illustration created in BioRender.com accessed on 17 October 2022].

**Table 1 cells-11-03768-t001:** Commonly used markers for different states of microglia in multiple sclerosis and specific animal models.

	Markers			
	Homeostatic Microglia	Classically Activated Microglia	Alternatively Active Microglia	Ref
Multiple sclerosis	TMEM119 CX3CR1 P2RY12 Iba1 * CSF1R	MRP14^+^ * TSPO * MHC II * CD68^+^ * CD74 * CD40 * CD86 * CD80 *	CD206 * TREM2 CD163 * CD309 * CCL22 *	[48,49,50,51,52,53,54]
Animal models				
Models that include adaptive immune mechanisms (e.g., EAE)	TMEM119 P2RY12 Iba1 * CX3CR1 CD11b^+^/CD45^−^ CSF1R CD68^−^	CD74 * CD40 * CD86 * CD80 * iNOS * TNF * IL-12 *	CD206 * TREM2 FIZZ1 * CD163 * Chil3 * Arginase 1 *	[53,55,56,57,58,59]
Innate immune toxin models (e.g., Cuprizone, lysolecithin)		TSPO * CD86 iNOS * MHC II * CCL2 * TNF * CD83 CD14	CD206 * Arginase 1 * TREM2 MRC1	[60,61,62,63,64]

* Also expressed by macrophages. CCL: CC chemokine ligand. CD: Cluster of differentiation. Chil3: Chitinase-like 3. CSF1R: Colony-stimulating factor 1 receptor. CX3CR1: C-X3-C motif chemokine receptor 1. FIZZ1: Found in inflammatory zone 1. Iba1: Ionized calcium-binding adaptor molecule 1. IL: Interleukin. iNOS: Inducible nitric oxide synthase. MHC II: Major histocompatibility complex. MRC1: Mannose receptor C-type 1. MRP14: Migration inhibitory factor-related protein 14. P2RY12: Purinergic receptor P2Y12. TMEM119: Transmembrane protein 119. TNF: Tumor necrosis factor. TREM2: Triggering receptor expressed on myeloid cells 2. TSPO: Translocator protein.

## Data Availability

Not applicable.

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
