# Peer review of "Modulation of the Microglial Nogo-A/NgR Signaling Pathway as a Therapeutic Target for Multiple Sclerosis"

_cells, 2022, doi:10.3390/cells11233768_

Round 1

Reviewer 1 Report

This review presents the pathological conditions of microglia in aging and MS in detail, which are interesting. However, there are some issues to be supplemented:

1.      The phenotype of microglia showed disease related characters, whether there are specific characters of microglia in MS?

2.      During the chronic stage of MS, microglia always showed immune tolerance status, which means it is failed to respond to the stimuli accompanied with their metabolic dysfunction. Please supplement it in the present review.

Author Response

  1. The phenotype of microglia showed disease related characters, whether there are specific characters of microglia in MS? Answer (A): We have now incorporated a Table that lists markers identified in MS and compared this with those of the commonly utilised animal models. We have now also included a paragraph from lines 208-225 to outline specific characteristics of microglia during chronic-active lesions in MS.
  2. During the chronic stage of MS, microglia always showed immune tolerance status, which means it is failed to respond to the stimuli accompanied with their metabolic dysfunction. Please supplement it in the present review. A): We have incorporated a paragraph from lines 208-225 to outline the neuroinflammatory signatures of microglia that are incorporated within chronic-active MS lesions and outline their metabolic signatures.

Author Response

Authors need to add some new and adequate references related to this topic such as:

  1. Guerrero BL, Sicotte NL. Microglia in Multiple Sclerosis: Friend or Foe?. Front Immunol. 2020;11:374. Published 2020 Mar 20. doi:10.3389/fimmu.2020.00374

  1. Absinta M, Maric D, Gharagozloo M, et al. A lymphocyte-microglia-astrocyte axis in chronic active multiple sclerosis. Nature. 2021;597(7878):709-714. doi:10.1038/s41586-021-03892-7

  1. Voet S, Prinz M, van Loo G. Microglia in Central Nervous System Inflammation and Multiple Sclerosis Pathology. Trends Mol Med. 2019;25(2):112-123. doi:10.1016/j.molmed.2018.11.005

Answer (A): The above-mentioned review articles have now been incorporated in the modified version of the manuscript as references 8, 72 and 11 respectively. Thorough discussion of the Absinta data has been incorporated in the new paragraph from lines 208 and 307, respectively.

-  Authors need to explain better practical implication of our new knowledge about microglia (clinical as well as therapeutic) in MS management. (A): The clinical and therapeutic focus of the current manuscript has been incorporated in the paragraph from line 563 and is only related to the title of the manuscript surrounding Nogo Receptor-dependent mechanisms. Further discussion of clinical/therapeutic intervention of microglia is beyond the scope of the current review and has been addressed by many recent review articles (example: Baecher-Allan C, Kaskow BJ, Weiner HL. Neuron. 2018 Feb 21;97(4):742-768.)

Reviewer 3 Report

The present review manuscript, which focuses on the microglial Nogo-A/NgR signaling pathway, has provided sufficient, detailed and up to date literature search of the relevant signaling pathway for targeting MS for its treatment.

Authors also provided considerable information regarding key properties of microglial cell function and its role in propagating neuroinflammation and neurodegeneration.

Few corrections and changes are needed as follows, which might improve the paper:

1.      Line 55-57: please discuss in detail and indicate the name of the specific neurotoxic factors (proinflammatory cytokines, chemokines).

2.      Line 58: As it has been explained briefly about the EAE model, it’s better to briefly explain about cuprizone-induced demyelination model and lysolecithin model too, as have been indicated throughout the manuscript.

3.      Please define abbreviations of the names of components illustrated in Fig.1 such as TRAF6, TRADD, NEMO, etc. in the caption of the figure.

4.      Line 652: the clinical trial for targeting the microglial Nogo-A/NgR signaling pathway has been indicated in the conclusion section. This could be odd and it’s recommended to displace it to a related section before the conclusion. Furthermore, more details are needed to be added in terms of this clinical study.

5.      In the conclusion section, it’s recommended not to use references and explain more briefly. The conclusion seems too long.

6.      At least one table is needed in this review article for assorting and summarizing the key points of the content, for instance experiments in targeting the microglia Nogo-A/NgR signaling pathway for mitigation of MS.

Author Response

Few corrections and changes are needed as follows, which might improve the paper:

  1. Line 55-57: please discuss in detail and indicate the name of the specific neurotoxic factors (proinflammatory cytokines, chemokines). Answer (A): This has now been modified to indicate all names for these acronyms in the modified version. Incorporation of a a reference for these factors has been added on lines 54 and 59 for the reader as the focus of this review is not related to the inflammatory signalling of these factors.
  2. Line 58: As it has been explained briefly about the EAE model, it’s better to briefly explain about cuprizone-induced demyelination model and lysolecithin model too, as have been indicated throughout the manuscript. (A): This has been incorporated in a Table with the comparison of the various animal models and MS.
  3. Please define abbreviations of the names of components illustrated in Fig.1 such as TRAF6, TRADD, NEMO, etc. in the caption of the figure. (A): This has now been amended in the Figure legend.
  4. Line 652: the clinical trial for targeting the microglial Nogo-A/NgR signaling pathway has been indicated in the conclusion section. This could be odd and it’s recommended to displace it to a related section before the conclusion. Furthermore, more details are needed to be added in terms of this clinical study. (A): This has now been moved as a paragraph commencing from line 563.
  5. In the conclusion section, it’s recommended not to use references and explain more briefly. The conclusion seems too long. (A): This has now been amended.
  6. At least one table is needed in this review article for assorting and summarizing the key points of the content, for instance experiments in targeting the microglia Nogo-A/NgR signaling pathway for mitigation of MS. (A): This has now been incorporated (see Table 1).